# Three-Dimensionally Printed Expandable Structural Electronics Via Multi-Material Printing Room-Temperature-Vulcanizing (RTV) Silicone/Silver Flake Composite and RTV

**DOI:** 10.3390/polym15092003

**Published:** 2023-04-23

**Authors:** Ju-Yong Lee, Min-Ha Oh, Joo-Hyeon Park, Se-Hun Kang, Seung-Kyun Kang

**Affiliations:** 1Department of Materials Science and Engineering, Seoul National University, Seoul 08826, Republic of Korea; seoulboy@snu.ac.kr (J.-Y.L.); ohminhaa@snu.ac.kr (M.-H.O.); pjhjudy@snu.ac.kr (J.-H.P.); rkdtpgns456@snu.ac.kr (S.-H.K.); 2Research Institute of Advanced Materials (RIAM), Seoul National University, Seoul 08826, Republic of Korea; 3Soft Foundry Nano Systems Institute (NSI), Seoul National University, Seoul 08826, Republic of Korea

**Keywords:** elastic soft material, 3D printing, negative Poisson’s ratio structure, conductive composite material

## Abstract

Three-dimensional (3D) printing has various applications in many fields, such as soft electronics, robotic systems, biomedical implants, and the recycling of thermoplastic composite materials. Three-dimensional printing, which was only previously available for prototyping, is currently evolving into a technology that can be utilized by integrating various materials into customized structures in a single step. Owing to the aforementioned advantages, multi-functional 3D objects or multi-material-designed 3D patterns can be fabricated. In this study, we designed and fabricated 3D-printed expandable structural electronics in a substrateless auxetic pattern that can be adapted to multi-dimensional deformation. The printability and electrical conductivity of a stretchable conductor (Ag-RTV composite) were optimized by incorporating a lubricant. The Ag-RTV and RTV were printed in the form of conducting voxels and frame voxels through multi-nozzle printing and were arranged in a negative Poisson’s ratio pattern with a missing rib structure, to realize an expandable passive component. In addition, the expandable structural electronics were embedded in a soft actuator via one-step printing, confirming the possibility of fabricating stable interconnections in expanding deformation via a missing rib pattern.

## 1. Introduction

Three-dimensional (3D) printing is a promising field of emerging technology for fabricating complex and customized structures [1,2]. Furthermore, it is used for research and commercial purposes for the recycling of thermoplastic composite materials [3,4] and small-quantity customized production, such as patient-specific tooth implants [5] and discontinued engine parts [6]. Although 3D printing can provide an on-demand manufacturing process, it is difficult to build multi-functional products due to its single-material-based architecture. So as to transform this into functional 3D customized structures with embedded features such as electronic sensing platforms, cell growth factors, structural healing factors, etc., there were many attempts to incorporate multi-functional materials in single objects such as conductors, elastomers, cells, and hydrogels [7,8,9]. Therefore, 3D printing is used for both the production of simple structures during their initial stages [10] and in various fields such as wearable electronics [11,12,13], soft actuators [14], and artificial biomedical implants [15]. Extensive research has been conducted on printing various functional materials for application in these fields [16,17]; these capabilities enable the creation of fully integrated and functional soft electronic devices in a simplified, single step [18].

Stretchable soft electronics can be applied in various fields, including wearable devices, biomedical applications, and robotic systems [19,20,21,22]. Stretchable materials, such as conducting polymers [23,24,25,26], ionic conductors [27], and liquid metals [28], can be used for conformal contact in interfaces across stretchable electronics applications. Alternatively, stretchable structures such as serpentine [29,30], mesh [31], longitudinal waves [32], and micro-crack methods [33] are utilized to incorporate rigid chips with stretchable components. In addition, reverse engineering methods [34], stimuli-responsive materials [35], and self-healing materials [36,37] have been incorporated to establish conformal contact with the biostructure. However, those strategies require the deformation of each component to achieve conformal contact with complex structures.

However, real-world applications require both conformal contact and stability during multidimensional deformation, which can be simply demonstrated by fabricating a metamaterial structure with a systematically negative Poisson’s ratio from an initial design [38,39,40,41,42]. However, regarding the manufacturing of electronic devices using pre-existing auxetic structures without 3D printing, two main complex approaches were used: (1) molding or coating the entire auxetic structure with conducting materials [43,44,45,46,47], and (2) fabricating a substrate of an auxetic structure, followed by patterning a conducting line on the top [48,49,50]. Method (1) has a lack of designing freedom for conducting parts in the structure, while method (2) provides a volumetrically inefficient architecture due to the stacking electronics, and adhesion should be thoroughly considered. The utilization of a 3D printer for multi-material printing can facilitate the segregation of both the structural and electronic components within a singular stereographic entity [51]. This novel approach also presents potential utility in the domain of structural electronics possessing a negative Poisson’s ratio.

For the 3D printable structural frame materials, RTV silicone was chosen because it is a widely commercialized silicone elastomer used for adhesive sealants, and various types for specialized purposes are already on the market; in addition, it already has viscoelastic properties that are directly applicable for 3D printing, with self-supporting characteristics [52]. In addition, for the conducting fillers, we considered Ag flakes because Ag is a noble metal that has higher conductivity than non-noble metals such as Cu or a carbon-based conductor such as carbon nanotubes and carbon black, while the flakes have a planar structure (2D) that can induce better percolation networks than a sphere-like structure (0D) or wire-like structure (1D) [47,51]. Although hydrogel is also a great candidate for soft electronics, the main difference between the two is the charge carrier, wherein a conductive filler/silicone composite uses electrons and a hydrogel uses ions [53]. Here, electronic components using electrons were more of interest for building passive components, such as the resistor, capacitor, and inductor used here (see Appendix A).

This study proposes the development of expandable voxelated auxetic structure-based stretchable soft electronics via a rigid room-temperature-vulcanizing silicone (rRTV) frame material and an elastic conductor (Ag-RTV) composite, consisting of silver (Ag) flakes and soft RTV (sRTV) silicone. A 3D multi-material printer can realize structural electronics based on the missing-rib auxetic structure without requiring a substrate. The electrically conducting trace with the auxetic structure maintains a constant resistance under 100% strain. Furthermore, a multi-nozzle 3D printing method can be used to produce an auxetic structure by combining Ag-RTV conductor material and rRTV structural frame materials, which can fabricate various passive component-based sensors, such as tactile sensors, strain sensors, and heaters. Furthermore, simplified multi-nozzle-based one-step 3D printing enables soft actuators, which can be embedded with various soft electronics, including a strain sensor and interconnection. Finally, surface-mounted devices (SMD) with soft electronics are integrated onto the soft actuators. These demonstrations indicate that various complex structures can be printed for specific uses and can be applied to extensive potential fields via their connectivity with various external devices.

## 2. Materials and Methods

### 2.1. Preparation of 3D-Printed RTV-Based Ink

RTV silicone (Permatex, IL, USA), Ag flakes (4-8 μm, Alfa Aesar, Haverhill, MA, USA), and 4-methyl-2-pentanone (TCI, Tokyo, Japan) were placed in a mixing container, based on the experimental conditions for the desired volume fractions. The mixture was placed in a mixer (ARE-310, THINKY, Tokyo, Japan) and evenly mixed, using the planetary centrifugal mixing mode, at a rotation speed of 2000 rpm for 4 min. The Ag-RTV mixture was used to fabricate the structure using a 3D printer (BIO X, Cellink, Gothenburg, Sweden). Thereafter, the resulting structure was placed in a 120 °C oven for 2 h to evaporate the lubricant. Finally, the production of the sample was completed by allowing the Ag-RTV to cure overnight in ambient conditions (~25 °C).

### 2.2. Characterization of 3D-Printed RTV-Based Inks and Electronic Components

MultiDrive rheometer (MCR 702e, AntonPaar, Graz, Austria) was used to discern the intricate rheological properties of the ink samples. The inks were carefully placed in a confined space between two plates with a precise gap of 0.5 mm. The sweep mode of the rheometer was used to accurately quantify the viscosity of the inks, covering a variety of shear rates from 0.1 to 100 s^−1^ at a constant temperature of 25 °C. Additionally, the amplitude mode of the instrument was used to assess the storage and loss shear modulus, encompassing a broad range of shear strain from 0.01 to 10% and an angular frequency of 10 rad/s. Samples for the feature size measurements were produced using a 3D printer (Cellink, BIO X, Sweden) and printing was performed at various speeds ranging from 1 mm/s to 6 mm/s at 70 kPa (RTV) and 60 kPa (Ag-RTV). The samples for the conductivity measurements were also fabricated via a 3D printer as aforementioned for feature size measurements, while resistance was measured using a digital multimeter (DMM). The conductivity was calculated using the measured resistance value and the dimension of the sample. To analyze the mechanical properties of the RTV and Ag-RTV composites, uniaxial tensile tests were executed using an Instron 3343 universal testing machine (Instron, Norwood, MA, USA) at a fixed receding strain rate of 100%/min. For the electromechanical properties measurements, as shown in Appendix A, samples were prepared by placing 3D-printed Ag-RTV on a Cu tape fragment, followed by multiple layers (10–15) of protection tape covering the Ag-RTV and Cu tape to prevent resistance changes in the contact area by jig. Then, cables were connected to both sides of the Cu tape and the cable was connected to the DMM. The strain changed from 0% to 100% in 5% intervals and resistance was measured at each fixed strain point with the DMM. An LCR meter was used to measure the capacitance of the sensors and the inductance of the printed inductor. Capacitance data were smoothed using the Origin program, with a percentile filter (points of window = 5, boundary condition = none, percentile = 65).

### 2.3. Multi-material Printing for Electronics and Actuators

The process of modeling the intricately shaped, bespoke components was meticulously executed using the computer aided design (CAD) software (Fusion 360, Autodesk, San Francisco, CA, USA). Thereafter, the 3D models were sliced by slicer program (Repetier-Host, Hot-World GmbH & Co. KG, Willich, Germany) which enabled the generation of a G-code that was customized to the specific shapes of the components. The G-code was carefully fine-tuned, which reduced the printing time. Furthermore, the finely tuned G-code was transferred to the 3D printer (BIOX, Cellink, Gothenburg, Sweden), which is equipped with three nozzles that are capable of simultaneously printing three different materials. The substrate was covered with polypropylene to prevent adhesion after curing (Appendix A). Speed and pressure were modified at various speeds ranging from 1 mm/s to 10 mm/s and at 60−100 kPa.

## 3. Results

### 3.1. Overview of 3D-Printed Expandable Structural Electronics

Figure 1 shows the overall research strategy, which includes the concepts and 3D-printable ink materials used for expandable structural electronics, along with the applications integrating pneumatic soft actuators. Figure 1a shows the multi-nozzle printing of soft RTV silicone (sRTV), rigid RTV silicone (rRTV), and Ag-RTV, highlighting their respective working principles as a frame and conductor. A hyper-elastic RTV silicone elastomer was utilized as the frame material, especially rRTV for the control frame in the structural electronics and sRTV for pneumatic actuators. RTV silicones involve a condensation reaction between a base polymer with reactive groups and a curing agent that acts as a catalyst to promote cross-linking at room temperature after printing, resulting in a cured silicone rubber material [54,55,56,57,58]. A stretchable conductor was used for the electron conduction pathway and was composed of Ag flakes and sRTV, with 4-methyl-2-pentanone as a lubricant. Here, 4-methyl-2-pentanone provided excellent dispersion of the Ag flakes in the sRTV, allowing excellent electron current flow through a percolation path of well-dispersed Ag flakes and adequate viscoelastic properties for 3D printing. Using rRTV and Ag-RTV within microstructures having a negative Poisson’s ratio provides stable electrical performance under expanding deformation. Figure 1b illustrates demonstrative examples, such as passive component-based tactile sensors, joint sensors, and heaters with 3D-printed expandable structural electronics applied to the surface of multi-dimensionally deforming areas on body parts such as the knee or elbow. Figure 1c shows the capability of expandable structural electronics to be integrated with an expanding body, which is a soft robotic system, in one-step multi-nozzle printing.

### 3.2. Characterization of 3D-Printable Soft Electronic Inks

To enable the implementation of soft electronics via 3D printing, optimization is required for both the electrical characteristics of the ink material as an electronic device and its rheological characteristics in order to produce a stacked structure. Figure 2 shows the characterization process, which includes material optimization and mechanical property testing for the application of sRTV and Ag-RTV ink materials via the 3D printer. Figure 2a and Appendix A show the real images of the printed 3D structures of RTV frame materials and Ag-RTV conductor materials. Commercial RTV silicone typically exhibits rheological properties that are suitable for 3D printing; thus, separate optimization is not essential. However, for Ag-RTV mixed with Ag flakes in sRTV, various optimizations are required to achieve the desired structure, as shown in Figure 2a. Figure 2b shows the addition of Ag flakes to the sRTV at various volume fractions for optimizing the conductivity of Ag-RTV. Furthermore, a percolation threshold was observed at a volume fraction of 17.5% Ag-RTV, confirming the rapidly increased conductivity. Furthermore, the higher the volume fraction of Ag flakes, the greater the percolation path, resulting in ~10^2^ S/cm conductivity, becoming saturated after a certain volume fraction (~30%). Figure 2c shows an experiment aimed at optimizing the volume fraction of lubricant to improve the printability of Ag-RTV while securing electrical conductivity. To aid the dispensing of Ag flakes in the sRTV silicone, 4-methyl-2-pentanone was used as a lubricant, which blends well in silicone elastomers [59]. Dispensed line structures, including Ag flakes and various 4-methyl-2-pentanones in sRTV, underwent lubricant vaporization at 120 °C, and the resulting conductivity was measured at room temperature. Ag-RTV with a volume ratio of 40% of lubricant showed excellent dispensing through the nozzle without clogging, and the dispensed line stacked well. Despite the increase in the lubricant-reduced conductivity of Ag-RTV, ~30 S/cm was reached at a 40% lubricant percentage, which is applicable for electronics. After the lubricant dried out, the ratio between the Ag flakes and sRTV became 31 vol% and 69 vol%, which showed good conductivity previously. Showing 3D printability with a securing conductivity condition for 3D printable Ag-RTV was determined as [Ag flake: sRTV: 4-methyl-2-pentanone = 18:40:42] in the volume fraction. Figure 2d,e shows the rheological properties of the optimized Ag-RTV, sRTV. The viscosity at a shear rate of 10s^−1^ was 210 Pas and 103 Pas, while the yield shear stress was 2.6 Pa and 1.01 Pa for Ag-RTV and sRTV, respectively, which showed a shear thinning behavior applicable for 3D printing under low pressure (< 100 kPa). Figure 2f displays the difference in feature size as a function of printing speed for the Ag-RTV and sRTV at the minimum dispensing pressure for each ink (Ag-RTV, 90 kPa; sRTV, 80 kPa) with a 27-gauge conical nozzle, and the printed line width was optimized to 300 μm for each ink. Figure 2g,h displays optical microscope (OM) images of the sRTV and Ag-RTV, respectively, at the same line width. As was evident from the images, both inks can be printed at a uniform line width. A tensile test was performed on 3D-printed Ag-RTV and sRTV samples. As shown in Figure 2i, the sRTV (blue) exhibited hyper-elastic mechanical properties with an elastic modulus of 83.59 kPa and a fracture occurring at a strain of approximately 650%. The Ag-RTV (red) exhibited a reduced elastic range of approximately 20% and a fracture occurring at approximately 350% strain, which was attributed to the reduced volume of sRTV and increased stress points due to the addition of the Ag flakes and lubricant. The evaporation of the lubricant also resulted in the formation of pores, which served as additional fracture points. SEM images of the Ag-RTV before and after tensile fracture revealed that the Ag flakes were uniformly dispersed in the ink, resulting in a degradation of the mechanical properties (Appendix A). Linear alignment and the partial agglomeration of Ag flakes were observed after fracture, indicating the migration of Ag flakes owing to external mechanical force. The rheological properties, the 3D-printed structure, and the mechanical properties of rRTV are further displayed in Appendix A.

### 3.3. A 3D-Printed Microstructure with a Negative Poisson’s Ratio for Expandable Soft Electronics

Figure 3 presents the negative Poisson’s ratio patterns for expandable structures that are produced via the 3D-printing of rRTV and calculates the Poisson’s ratio of the expandable structure, based on the strain changes. The geometrical determination of the unit cell of the structure, both missing-rib and modified missing-rib forms, is shown in Appendix A. The distance (r) of the pattern was 2.121 mm and the pattern feature size was 0.4 mm. Figure 3a shows the three types of expandable structure arrays, which are composed of simple unit cells. Each pattern type was photographed using a DSLR camera equipped with a macro-lens, with the structure changing as the engineering strain increased from 0 to 20% along the *x*-axis at 2.5% intervals. A marker was placed at the 1/4, 2/4, and 3/4 positions on the *y*-axis to enable the calculation of the change in engineering strain using Image J software (National Institutes of Health, Bethesda, USA), while the average and standard deviation are displayed on a graph (Figure 3b). Figure 3c indicates the Poisson’s ratio changes of each of the three types of expandable structures, shown according to the strain change. The Poisson’s ratio of pattern type 1 changed from −0.1448 (2.5%) to −0.0185 (20%), while for pattern type 2, it appeared to be maintained at −0.3. The Poisson’s ratio of pattern type 3 decreased overall, with the absolute value of the Poisson’s ratio increasing from −0.452 (2.5%) to −0.6 (15%) and converging to −0.457 (20%). When sRTV and rRTV were used, respectively, in pattern type 1 to measure the Poisson’s ratio with strain dependency, the Poisson’s ratio had a positive value (an average of ~0.05), according to the vertical strain in the case of sRTV (Appendix A). Therefore, the spring constant of sRTV for the auxetic effect was not appropriate, while the moduli of Ag-RTV and rRTV were similar and could provide auxetic effects (Ag-RTV, 708.2 kPa, and rRTV, 977.1 kPa).

### 3.4. Three-Dimensionally Printed Control Frames for Expandable Soft Electronics and Their Applications

In the context of printing an expandable structure, variable expandable electronics were enabled by the partially printed conductor (Ag-RTV). Figure 4 shows an example of an expandable soft electronic device application. Figure 4a,b shows images of the printed conductor, where Figure 4a shows a non-structured printed resistor and Figure 4b displays an expandable-structure-based resistor. Figure 4c shows the resistance change of two different resistors when the strain was applied from 0 to 100% (non-structured is in red; expandable-structured is in blue). Regarding non-structured resistors, resistance continued to increase with the strain. After 60% of strain, a rapid increase in resistance was observed owing to the decreased percolation path of the Ag flakes as the polymer was stretched. Regarding the expandable-structured resistor, the change in resistance was insignificant despite 100% strain being applied. Furthermore, the stretch deformation of the polymer could be reduced. Figure 4d shows a photograph (left) and a magnified view (right) of a capacitor-type strain sensor, based on a structure wherein RTV forms the structure on the *x*-axis and Ag-RTV forms the electrode on the *y*-axis. Figure 4e presents the measured capacitance values of the sensor under different applied pressures. The capacitance remained at 6 pF under no external force but decreased to approximately 4 pF when pressed with one finger and decreased to 2 pF when pressed with two fingers, demonstrating the sensor’s variable pressure sensitivity.

Figure 4f shows an image of an electronic device attached to the knee to measure the capacitance change according to the strain. Figure 4g shows the data measuring the capacitance change according to the leg angle change after attachment. A capacitance of approximately 6 pF was maintained when the leg was not moved, while a capacitance of approximately 20 pF was confirmed when the leg angle was changed by approximately 45°. A large change in the leg angle of approximately 90° resulted in an increased capacitance of 20 pF. The distance between the two electrodes forming the capacitor and the conformal contact of the fabric increased according to the knee movement, which increased the capacitance. Figure 4h shows a photograph of an expandable-structured resistor-type heater. The expandable structure could function stably as an electronic device because of the negligible change in resistance to 100% strain. Figure 4i shows the IR image when the heater is turned on/off. When the heater is off (left), the overall temperature is between 20 and 30 °C. However, when the heater was operated under 1 MHz and 30 Vpp conditions (right), the overall temperature of the heater was between 50 and 60 °C, with a local temperature of up to 80 °C. This confirmed the operability and applicability of deformable electronics using the 3D-printing fabrication method. In addition, the inductor, which is an essential passive electronic component for wireless operation or wireless power transfer, was printed using Ag-RTV and sRTV and showed a resistance of 128 Ω and an inductance of 73.5 μH at 1 MHz, with 0.136 Q (Appendix A).

### 3.5. Fully 3D-Printed Soft Actuators Embedded in Expandable Soft Electronics

Figure 5 shows the fabrication process and presents various application demonstrations highlighting the potential of one-step manufacturing to simultaneously create a 3D-structured soft actuator and a soft electronic device. Figure 5a shows an exploded view schematic of the 3D-printed actuators and sensors, which have been integrated into the strain limiter of the soft actuator. The pneumatic actuator was fabricated using sRTV, which is highly deformable, with a thick bottom designed to function as a strain limiter. Appendix A shows other possible structures for soft pneumatic actuators, such as a triangular (1-channel) or hexagonal (3-channel) structure. For the control frame, deformation occurs in a controlled direction using rRTV, while for the soft electronics, Ag-RTV was used to facilitate electronic operation. Figure 5b shows a photograph of an actuator integrated with electronics fabricated using a 3D printing method. Two types of electronics were integrated into the actuator: a non-structured, resistor-based strain sensor composed of Ag-RTV (left) and an expandable-structured, deformation-resistive electrode composed of rRTV and Ag-RTV (right). Figure 5c,d displays the resistance changes in the embedded electronics during actuator operation.

Figure 5c shows data from the non-structured, resistor-based strain sensor, revealing a 1.1-fold increase in resistance during pneumatic actuation, compared to initial resistance. This confirms the sensor’s potential for use as a strain sensor. The data also show that the resistance increases as the actuation is repeated and a strain beyond the elastic range of Ag-sRTV is applied. In the same structured actuator, when pneumatic actuation was applied, a similar bending angle showed about a 20% strain variation [60]. In Appendix A, a sharp increase in resistance is observed at a 20% strain for the non-structured conductor. Microscopically, as seen in the SEM image of Appendix A, the Ag flakes align in the direction of the strain, which creates gaps between them and leads to a decrease in the percolation path. This result causes a sharp increase in resistance when the actuation occurs. Figure 5d shows data measuring the change in electrode resistance during the pneumatic deformation of the actuator when embedded with an expandable-structured electrode. Furthermore, the electrode’s resistance did not change, even after multiple actuations were performed. This demonstrates the potential use of the electrode as an electronic device by structurally suppressing the material’s intrinsic resistance. Figure 5e shows a photograph of a surface-mounted device (SMD) chip being operated through the embedded stretchable conductor of a fabricated SOF actuator. Therefore, various and complex structures can be printed for specific uses and can be extended to potential fields of application via connectivity with external devices, such as SMD chips.

## 4. Discussion

In this study, we proposed three types of 3D printable ink materials for expandable structural electronics, with three types of missing-rib structures. Firstly, for 3D printable materials, Ag-RTV is used as a stretchable conductor, with high-strength elastic rRTV as a control frame and hyper-elastic sRTV as an object frame for the pneumatic actuator in this case. Secondly, for the structures, other patterns possessing negative Poisson’s ratio can be used in expandable electronics to prevent changes in the cross-section area of the conductor, which may result from the opening of cells in the pattern [47]. Patterns that possess a negative Poisson’s ratio, such as re-entrant, rotational, chiral, and crumpled sheets, can be used for this purpose. The missing-rib structure was chosen because it divided the pattern into rows and columns and allowed for its systematic design and modification [61,62,63,64]. We demonstrated the fabrication of expandable structural electronics via multi-material printing, such as capacitor-type strain sensors and resistor-type heaters. Our one-step 3D-printing process simplified the manufacturing system and enabled the realization of soft actuators that can be embedded in various structured electronics. Our findings show the potential of these 3D printable ink materials and expandable structures for the development of soft and stretchable electronics.

## 5. Conclusions

▪Merits:
-Expandable structural electronics, in the form of metamaterial structures (missing-rib formations) were made possible by configuring voxelated elastic conductors and frame materials via multi-material printing.-A modified missing-rib-structured conductor with a control frame (Ag-RTV/RTV) showed stable resistance up to a 100% strain change.-A stereographically designed missing-rib-structured tactile sensor, strain sensor, heater, and monolithically integrated sensors/interconnection embedded pneumatic soft actuator were demonstrated.▪Limitations:
-More investigations are needed to increase the deformation range where the resistance is stable.-Various cell-opening auxetic structures can be examined for better negative Poisson’s ratios.-More studies should be conducted into the reliability of Ag-RTV, not only in terms of deformations but also in terms of chemical, humidity, and other real-life factors.▪Future scope:
-More complex 3D auxetic structures, consisting of conducting voxels and frame voxels, can be explored for various applications.-Customized soft electronics with actuators can be demonstrated in cases requiring multi-dimensional deformations, for use in automobiles, aerospace, fashions, etc.

## Figures and Tables

**Figure 1 polymers-15-02003-f001:**
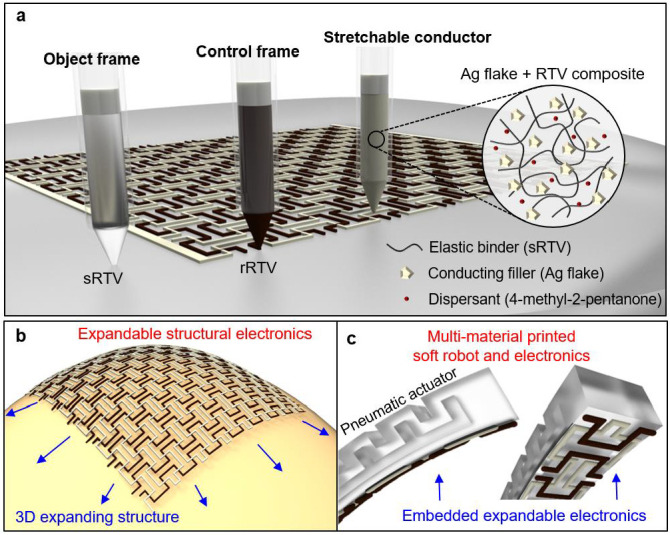
The overall strategy of 3D-printed soft electronics via multi-nozzle printing fabrication. (**a**) Scheme of the ink components for 3D-printed hyper-elastic soft materials. (**b**) Illustration of expandable structural soft electronics application to a 3D expandable structure. (**c**) Scheme of a multi-material one-step 3D-printed soft actuator with embedded soft electronics.

**Figure 2 polymers-15-02003-f002:**
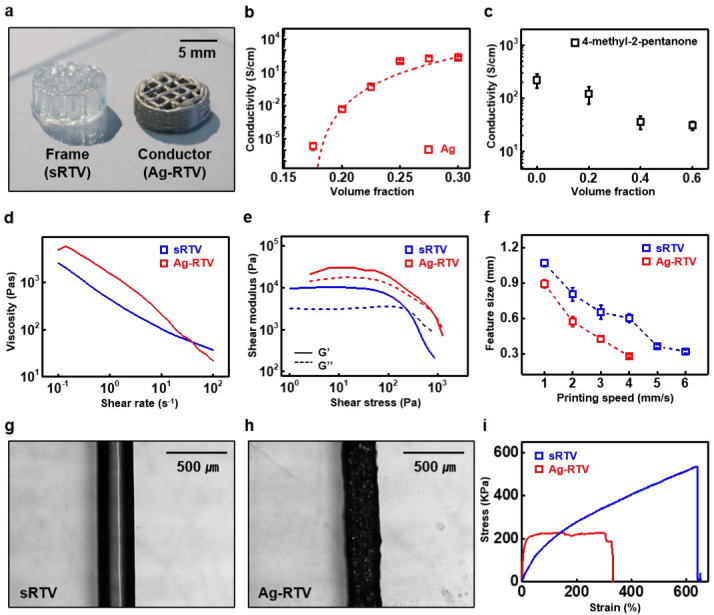
Characterization of ink materials. (**a**) Real images of the 3D-printed structure using RTV (left) and Ag-RTV (right) inks. (**b**) Conductivity changes of Ag-RTV with various volume fractions of Ag flakes. (**c**) Conductivity changes of Ag-RTV with various volume fractions of lubricant (methyl-pentanone). (**d**) Plot of the apparent viscosity versus shear rate for RTV (blue) and Ag-RTV (red) inks. (**e**) Comparison of the storage modulus G′ (solid line) and loss modulus G″ (dashed line) as a function of shear stress for RTV (blue) and Ag-RTV (red). (**f**) Experimental measurements of the printing resolution according to the printing speed (RTV, blue; Ag-RTV, red). (**g**,**h**) Line width of the optical microscopic image of (**g**) RTV and (**h**) Ag-RTV. (**i**) Stress-strain behavior of the RTV (red) and Ag-RTV composites (blue).

**Figure 3 polymers-15-02003-f003:**
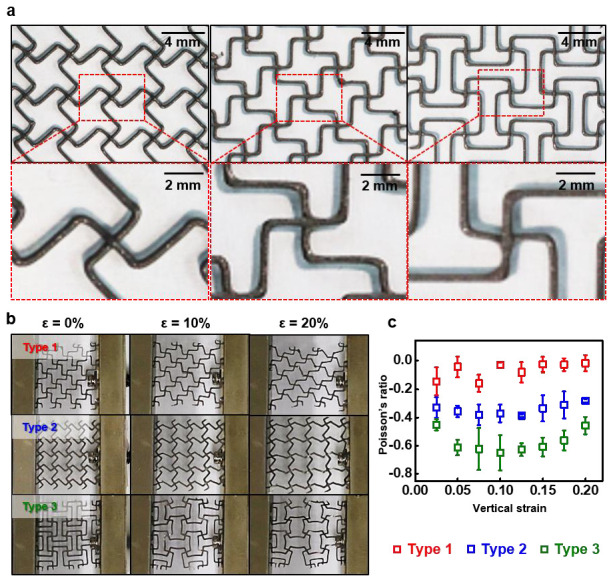
A 3D-printed negative Poisson’s structure using RTV silicone. (**a**) Photographs of various 3D-printed negative Poisson’s structures using RTV silicone. Different types of missing-rib structures for types 1 to 3 with a line width of 0.4 mm. (**b**) Real images of the RTV negative Poisson’s structure deformation with various strains (left, 0%; center, 10%; right, 20%). (**c**) Poisson’s ratio calculation graph for the various structures.

**Figure 4 polymers-15-02003-f004:**
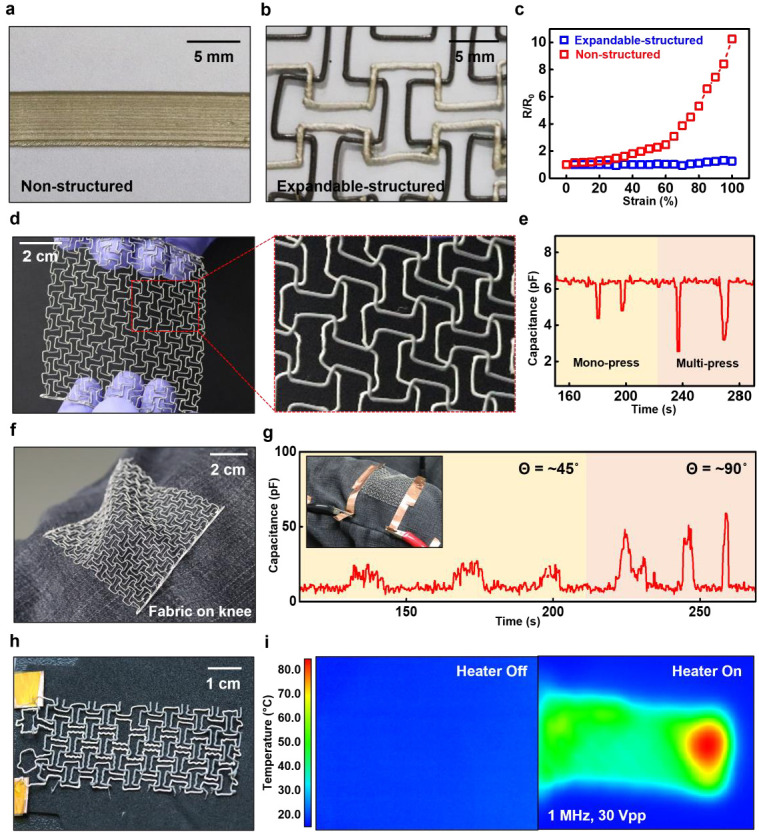
A 3D-printed negative Poisson’s structure via Ag-RTV silicone. (**a**) Photographs of a 3D-printed non-structured sample using Ag-RTV silicone. (**b**) Photographs of a 3D-printed negative Poisson’s auxetic sample using RTV and Ag-RTV silicone. (**c**) Plot of resistance change (R/R_0_) versus strain (non-structured is in red; auxetic-structured is in blue). (**d**) Real images of an auxetic-structured capacitor-typed strain sensor array (full view, left; enlarged view, right). (**e**) Capacitance changes of sensor arrays, depending on the pressure. (**f**) Image of an auxetic-structured capacitor-type strain sensor array with conformal contact on the knee. (**g**) Capacitance changes of the sensor arrays according to the various knee angles. (**h**) Photographs of a 3D-printed auxetic-structured resistor-based heater on the fabric. (**i**) Temperature variations of a 3D-printed resistive heater measured by an infrared (IR) camera (heater off, left; heater on, right).

**Figure 5 polymers-15-02003-f005:**
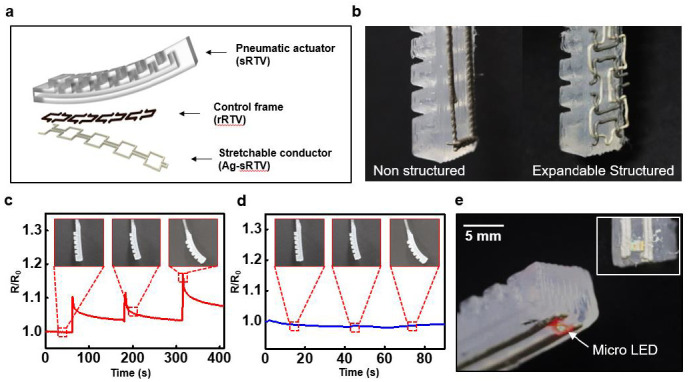
Fully 3D-printed perceptive soft actuator and electronics, via RTV and Ag-RTV. (**a**) Scheme of the inner structure of a 3D-printed soft actuator using hyper-elastic RTV and Ag-RTV. (**b**) Scheme of the exploded view of soft actuators and embedded electronics with various structures. (**c**) Real images of micro-LED operation using a 3D-printed soft actuator and electronics. (**d**) Resistance change (R/R_0_) of non-structured resistor-based strain sensors versus the actuation of a 3D-printed soft actuator. (**e**) Resistance change (R/R_0_) of an auxetic-structured resistor-based strain sensor versus the actuation of a 3D-printed soft actuator.

## Data Availability

The data presented in this study are available on request from the corresponding author.

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
