# Peer review of "Three-Dimensionally Printed Expandable Structural Electronics Via Multi-Material Printing Room-Temperature-Vulcanizing (RTV) Silicone/Silver Flake Composite and RTV"

_polymers, 2023, doi:10.3390/polym15092003_

Round 1
Reviewer 1 Report
In this manuscript, the authors designed 3D-printed expandable structural electronics in a substrate-less auxetic pattern that can be adapted to multi-dimensional deformation. Ag-RTV and RTV were printed as conducting voxels and frame voxels. The expandable structural electronics were embedded in a soft actuator via one-step printing. I suggest the acceptance of this manuscript in polymers if the author can address the following concerns.
1. In section 3.2, “Therefore, the optimized condition for 3D printable Ag-RTV was [Ag flake : sRTV : 4-176 methyl-2-pentanone = 18: 40: 42] in volume fraction.” 40% of sRTV was explained. How about 18% of Ag flake and 42% of 4-176 methyl-2-pentanone? Why these numbers were optimized values?
2. In Figure 2(i), sRTV with lubricant has higher strength and elongation. Usually, lubricant will decrease the strength and increase the elongation. Particles usually decrease elongation and increase strength. Why sRTV has a higher strength and elongation compared to Ag-RTV?
3. The author proposed a silicone composite for flexible electronics via 3D printing. Likewise, 3D printing of hydrogels (DOI: 10.1002/adfm.202107437), another kind of soft material, was also promising for stretchable electronics. The author can compare the difference and similarities between 3D-printed elastomers and hydrogels.
4. In Figure 3(c), the three types of patterns show a different trend when the vertical strain changes. What is the reason for this? Is there some experimental error? If it has, please show the error bar.
5. In Figure 4(e), “The capacitance remained at 6 pF under no external force but decreased to approximately 4 pF when pressed with one finger and 2 pF when pressed with two fingers, demonstrating the sensor's variable pressure sensitivity.” How to qualify for the pressure? “One finger” and “two fingers” represented about how many newtons?
6. The authors show the optical microscopic image of (g) RTV and (h) Ag-RTV. Can authors provide an image of sRTV? Is there any exterior difference between RTV and sRTV?

The author should try to improve the English quality to meet the high standard.
Reviewer 2 Report
It is an original paper dealing with “3D-printed expandable structural electronics via multi-material printing room-temperature-vulcanizing (RTV) silicone/silver flake composite and RTV “.Regarding this manuscript there are some minor and major comments below to help the readers to be more beneficial from the paper.
1. In abstract the authors say “Three-dimensional (3D) printing has various applications in many fields, such as soft electronics, robotic systems, and biomedical implants. “There is another important application. It is recycling of thermoplastic composite materials. It should be indicated in the abstract.
2. In introduction, line 32, the author described several commercial purposes of material extrusion 3D printing. There is another example and it is recycling of industrial thermoplastic composite materials. The authors can address to the following references
[a] Recovery of Particle Reinforced Composite 3D Printing Filament from Recycled Industrial Polypropylene and Glass Fibre Waste, Proceedings of the World Congress on Mechanical, Chemical, and Material Engineering, (MCM'22). Prague, Czech Republic; 2022.
[b] Manufacture of Composite Filament for 3D Printing from Short Glass Fibres and Recycled High-Density Polypropylene, Proceedings of the World Congress on Mechanical, Chemical, and Material Engineering,, Chemical, and Material Engineering (MCM'22). Prague; 2022.
3. In introduction line 43, for the stretchable conductive polymer for electronic application refer to the references below
[c] Piezoresistive pressure sensor based on conjugated polymer framework for pedometer and smart tactile glove applications. Sensors and Actuators A: Physical, 350, 114139
[d] MWCNT–epoxy nanocomposite sensors for structural health monitoring. Electronics, 7(8), 143.
[e] Polymer nanocomposite meshes for flexible electronic devices. Progress in Polymer Science, 107, 101279.
4. In section 2.2, line 103, the authors claim that they could measure the increase in electrical resistance by mustimeter, but neither electrical data recorded nor electromechanical setup presented.
5. What is the sharp increase in the electrical diagram in Fig.5, How do these sharp increases relate to the behavior of 3D printed materials?
6. . I believe the conclusions instead of discussion in the last section. Use bullets in Conclusions to emphasise the main achievements of the paper
Minor editing of English language required
Reviewer 3 Report
J. -Y. Lee et. al., submitted the paper entitled “3D-printed expandable structural electronics via multi-material printing room-temperature-vulcanizing (RTV) silicone/silver flake composite and RTV” to publish in “Polymers (I.F= 4.967)”. In this manuscript, author describes the use of 3D printed room-temperature-vulcanizing (RTV) silicone/silver flake composite in electronics as future technology. This work is an impressive one, can be accepted after addressing the queries.
1. In the introduction deliver a justification for the selection of RTV silicone and Silver flakes in this report (Especially Ag flakes). Also, it is essential to cite the relevant literature.
2. Sections 3.2 to 3.5 looks like mixture of results and discussion. Author must split the discussion part into discussion section. Currently, this is confusing with respect to Journal’s format.
3. Figure 3C, for Poisson’s ratio calculation on 0%, 10% and 20%, how many sets of data points considered? (At least 3-sets must be considered for discussion).
4. Figure 4g possess any noise peaks? It is advisable to deliver the smoothened capacitance data.
5. Discussion section is an incomplete section and should be boosted in the revised version.
6. Deliver the conclusion/summary section with merits, limitations and future scope.
7. Cite the following references for RTV silicone: 1. Energies 2017, 10, 1054; 2. Coatings 2021, 11, 312; 3. Polymers 2019, 11, 1142; 4. Polymers 2023, 15, 1224 and 5. Mater. Res. Express 9 (2022) 045304.
English looks fine, but requires moderate editing
Reviewer 4 Report
The paper "3D-printed expandable structural electronics via multi-material printing room-temperature-vulcanizing (RTV) silicone/silver flake composite and RTV" by Lee et al demonstrates a novel one-step 3D printing technology that yield expandable electronic structures for biomedical applications, especially in the field of wearable sensors.
This topic is of special interest to broad scientific community as current wearable electronic devices are cumbersome in their manufacture processes, which typically invovle microfabrications. A single step 3D printing step that yield these sensors will drastically improve their applicability in daily life. The paper owns great quality with clear demonstrations, reasonings, experimental details, and conclusions. I highly suggest to accept this paper after several minor modifications.
1. There should be no period in title (line 4)
2. The introduction and abstract need refinement. Audience without sufficient background will get confused by the main point of the paper. Is it a new 3D printing process? Is it a new 3D printing material? Or is it an old process and old materials but combined in new applications. I suggest the authors to start from the need of current biomedical applications and limitations of current 3D printing technology. Then clearly illustrate the new strategy that authors adopt to solve these dilemma, emphasizing the innovative points.
The english is clear and concise. Please eliminate the period mark in title.
Round 2
Reviewer 1 Report
I suggest the acceptance of this manuscript after the revision.
The author should try to improve the language quality before online publication.
Reviewer 2 Report
The paper is accepted from my side.